# The Impact of Innate Components on Viral Pathogenesis in the Neurotropic Coronavirus Encephalomyelitis Mouse Model

**DOI:** 10.3390/v15122400

**Published:** 2023-12-09

**Authors:** Brendan T. Boylan, Mihyun Hwang, Cornelia C. Bergmann

**Affiliations:** 1Department of Neurosciences, Lerner Research Institute, Cleveland Clinic, Cleveland, OH 44196, USA; boylanb@ccf.org (B.T.B.); hwangm@ccf.org (M.H.); 2Department of Pathology, Case Western Reserve University, Cleveland, OH 44106, USA; 3Molecular Medicine, Cleveland Clinic Lerner College of Medicine, Case Western Reserve University, Cleveland, OH 44106, USA; 4Department of Biological, Geological and Environmental Sciences, Cleveland State University, Cleveland, OH 44115, USA; 5School of Biological Sciences, Kent State University, Kent, OH 44242, USA

**Keywords:** murine hepatitis virus, neurotropic coronavirus, encephalomyelitis, interferons, interferon stimulated genes, pattern recognition receptors, meninges, T cells

## Abstract

Recognition of viruses invading the central nervous system (CNS) by pattern recognition receptors (PRRs) is crucial to elicit early innate responses that stem dissemination. These innate responses comprise both type I interferon (IFN-I)-mediated defenses as well as signals recruiting leukocytes to control the infection. Focusing on insights from the neurotropic mouse CoV model, this review discusses how early IFN-I, fibroblast, and myeloid signals can influence protective anti-viral adaptive responses. Emphasis is placed on three main areas: the importance of coordinating the distinct capacities of resident CNS cells to induce and respond to IFN-I, the effects of select IFN-stimulated genes (ISGs) on host immune responses versus viral control, and the contribution of fibroblast activation and myeloid cells in aiding the access of T cells to the parenchyma. By unraveling how the dysregulation of early innate components influences adaptive immunity and viral control, this review illustrates the combined effort of resident CNS cells to achieve viral control.

## 1. Introduction

Viral infections of the CNS are rare but can be fatal or have devastating long-term sequelae [1,2,3,4,5]. The recognition of invading viruses by pattern recognition receptors (PRR) is essential to set off a cascade of ‘alarm bells’ which alert resident cells to set up innate defense responses and recruit leukocytes to control the infection. PRR signaling activates pathways regulated by interferon regulatory factors (IRFs), NF-κB, and mitogen-activated protein kinases (MAPKs) [6,7]. IRF activation mediates the production of type I interferons (IFN-I), which in turn trigger signaling through the IFN-I receptor (IFNAR) to induce transcription of greater than 300 IFN-stimulated genes (ISGs). IFN-I in mice comprises a single IFNβ, 13 IFNα subtypes, and several other single-gene products [8]; this review focuses only on IFNα/β. ISGs encode a wide variety of proteins that can interfere at multiple levels with viral replication as well as cellular functions [9,10]. In addition to anti-viral factors, ISGs encode PRRs and signaling components of the IFNα/β pathway, thereby elevating an ‘anti-viral state’ in both infected and uninfected cells [11]. This amplification loop is especially relevant in vivo to better sensitize infected cells to infection and increase their anti-viral arsenal. ISGs also comprise genes in the major histocompatibility complex (MHC) antigen presentation pathway, thereby linking innate responses to adaptive T-cell immunity [10,12,13]. While many ISGs thus benefit the host by restraining viral dissemination, the effects on cellular functions can be toxic, especially to CNS cells, and result in apoptosis [14,15]. To limit such detrimental responses, IFN signaling also activates negative regulatory factors [10].

While this response pattern applies broadly to all infections, the diversity and magnitude of the anti-viral response is dependent on the signaling cascade induced by the relevant PRRs engaged, as well as the nature of the viral genome and replication cycle. It is also critical to note that the repertoire and levels of basal PRRs, as well as genes in the IFNα/β pathways, can differ vastly amongst cell types, thereby eliciting cell-type and organ-specific responses to any given infection. The complex regulation of PPR and IFN signaling has recently been reviewed [6]. Irrespectively, virtually all nucleated cells, including resident CNS cells, have the capacity to induce and respond to IFNα/β as primary anti-viral defense mechanisms [16,17,18,19,20]. The high susceptibility of IFNAR-deficient (IFNAR^−/−^) mice to neurotropic viruses underscores the essential role of IFNAR in protecting the CNS [21,22,23]. Keeping cell-type-specific responses in mind, this review will specifically highlight insights gained from the neurotropic murine coronavirus (mCoV) model on how early innate events induced in the CNS can shape the adaptive response and viral control. Special consideration is given to the following areas: (1) the cell type-dependent role of IFN signaling in controlling viral dissemination, (2) the anti-viral and immune modulatory roles of select ISGs, and (3) the interface of perivascular/meningeal immune events and parenchymal T cell.

### 1.1. Distinct Features of Different CNS Cell Types as Sensors and Responders to Infection

Neurotropic viruses enter the host from the periphery and can access the CNS by invasion from the blood as cell-free virus or via infected cells, transmission from peripheral nerves into the spinal cord via motor or sensory neurons, and/or through infection of neurons in the olfactory epithelium and spread along the olfactory nerve [5,7]. Once in the CNS, resident cells rely on the activation of PRRs to initiate IFNα/β and proinflammatory responses. It is becoming increasingly clear that CNS resident glia, neurons, and vascular cells not only display cell-type-specific but also region-dependent patterns of gene expression involved in innate signaling, which all contribute to the outcome of infection [16,17,20,23,24,25]. Aside from anti-viral functions, the importance of low levels of homeostatic IFNα/β production in brain function is evident from genetic CNS diseases associated with a dysregulated IFNα/β pathway [26,27,28,29]. The sensitivity of the CNS function to IFNs emphasizes the importance of balancing responses to viral CNS infections while minimizing pathological consequences.

Microglia, the resident macrophages of the CNS, are the primary CNS resident cells sensing homeostatic disruption caused by infection or non-infectious insults, including metabolic changes [1,30,31]. This is readily observed by dynamic morphological changes from a highly ramified state to a thickening of processes and an extension to sites of injury. Microglia and astrocytes express the broadest array of PRRs and are implicated as primary inducers of IFNα/β [1,20]. The differences in both the range and magnitude of basal and inducible transcripts encoding PRRs and factors in the IFNα/β pathway in CNS cell types further demonstrate the dynamic complexity and interdependence of CNS cells in optimizing immunity against viral infections. The nature of viral tropism, its replication cycle as well as the arsenal of genes capable of interfering with immune defenses add another level of complexity. As reviewed below, the mCoV model has provided extensive insights into these complex interactions by analyzing how a deficiency in several innate molecules and cell types can dysregulate adaptive responses.

### 1.2. The mCoV Encephalomyelitis Model

Infections by neurotropic mouse hepatitis viruses (MHV), belonging to the beta *coronaviridae* family, have provided numerous insights into the viral and host determinants contributing to viral tropism, dissemination, immune control, persistence, and demyelinating pathology [32,33,34,35,36]. Sublethal MHV strains commonly used to study the interaction of CNS resident cells and immune cells in viral pathogenesis are the MHV-A59 strain, causing mild encephalomyelitis and moderate hepatitis [33], and a monoclonal antibody-derived neutralization escape variant derived from the highly neurovirulent MHV-JHM strain, designated JHM v2.2-1 [37]. Both virus recombinants and cDNA clones have facilitated the characterization of viral determinants contributing to immune modulation and tracking infected cells via chemokine, enhanced green fluorescent protein (eGFP), or Cre recombinase expression [38,39,40,41,42]. To study MHV neuropathogenesis, virus can be administered intracranially (i.c.) or via the intranasal (i.n.) route. The latter requires 10–100-fold higher doses and establishes a more robust infection in peripheral organs, especially with the dual hepato- and neurotropic MHV-A59.

MHV-JHM v2.2-1 is predominantly glia tropic infecting few neurons, distinct from parental MHV-JHM, which infects both glia and neurons. MHV-A59 also infects both glia and neurons but causes milder neurological clinical disease. Common traits of both viruses are the establishment of persistent CNS infection in spinal cord white matter tracts characterized by the detection of low levels of persisting viral RNA in the absence of infectious virus and immune-mediated white matter demyelination [32,36]. Like all CoVs, MHVs induce modest IFNα/β expression and counteract the IFNα/β pathway. Nevertheless, IFNα/β responses are essential to limit viral dissemination and prevent mortality [43,44,45,46]. Early cellular infiltrates are neutrophils, monocytes, and NK cells. CD8 T cells and CD4 T cells are essential to reducing infectious virus below detection levels using both perforin- and IFNγ-mediated mechanisms. Nevertheless, low viral RNA levels persist in spinal cords, and viral re-emergence is controlled by local antibody-secreting cells [32]. It is unclear how viral RNA persists, yet it is associated with ongoing smoldering immune activation [32,47].

For the purposes of this review, the innate response in the CNS is broken down into three main aspects: the importance of IFNAR signaling on CNS resident cells for limiting virus dissemination and optimizing IFNγ signaling, the role of individual innate signaling molecules and ISGs in dysregulating immune functions, and the contribution of stromal and myeloid cells in promoting T cells access into the parenchyma.

## 2. Interplay between IFNα/β Inducer and Responder Cells in Controlling Viral Dissemination

The cell-type-specific differences in the ‘sentinel’ function of pathogens necessitate the coordination between IFNα/β inducers and responders to optimize the host’s anti-viral defense. Cervantes-Barragan et al. assessed relevant IFNα/β inducer and IFNα/β responder cell types critical to limit MHV-A59 dissemination following either systemic intraperitoneal (i.p.) or peripheral i.n. infection [43,48]. While plasmacytoid dendritic cells (pDCs) were revealed as vital IFNα/β-producing cells, IFNAR expression on myeloid cells was necessary for the early control of peripheral replication. Mice conditionally deficient for IFNAR further demonstrated that LysM^+^ macrophages and CD11c^+^ DCs, but not CD4^+^ T or CD19^+^ B cells, were most critical to contain MHV-A59 and prevent rapid lethal liver disease following systemic infection. IFNAR expression by macrophages was also deemed most important in controlling MHV in peripheral tissues following i.n. infection but had no effect on replication in the brain. The reliance on IFNα/β-mediated crosstalk between pDCs and responding macrophages and conventional DCs in preventing fatal infection of peripheral organs, but not the brain, highlights the context-dependent and organ-specific importance of IFNAR expression. Notably, upon i.n. vesicular stomatitis virus (VSV) infection, microglia become activated and establish an early innate barrier in the olfactory bulb to prevent viral spread within the CNS. Surprisingly, however, this was independent of IFNAR signaling by microglia, but it was reliant on IFNAR signaling by neurons and astrocytes to activate microglia [19].

Following CNS, i.c. infection with MHV-A59 or JHM v2.2-1, *Ifnα/β* mRNA peaked between days 3 and 5 post-infection (p.i.) [25,44,46,49,50]. Microglia appear to be the primary cell population inducing IFNα/β using the intracellular viral sensor, melanoma differentiation-associated protein 5 (MDA5) [25,46,51]. This is attributed to their elevated basal transcript levels of PRRs, including MDA5, as well as their higher constitutive expression of IFNβ and ISG compared to other glia and neurons [20,46]. The expression analysis of genes associated with the IFNα/β pathways in microglia and oligodendrocytes sort purified from MHV-JHM v2.2-1-infected mice further revealed that PRRs and ISGs were upregulated in both cell types, but the repertoire and transcript levels were more limited in oligodendrocytes [25]. Furthermore, although oligodendrocytes harbored elevated viral RNA compared to microglia, they never induced *Ifnα/β* mRNA. Even stimulation with the double-stranded RNA analog poly I:C failed to induce *Ifnα/β* mRNA in oligodendrocytes. The overall limited anti-viral response by oligodendrocytes was associated with delayed and sparse upregulation of *Ikkε* and *Irf7* transcripts, both key to amplifying IFNα/β responses. These results strongly supported that oligodendrocytes from the adult CNS are poor sensors of viral RNA, relying on exogenous IFNα/β to establish an anti-viral state, albeit with a limited array of anti-viral proteins. Additionally, their dependence on IFNγ to upregulate major histocompatibility complex class I (MHC-I) may contribute to their susceptibility to acute and persistent infection due to reduced CD8 T cell recognition [49,52].

A similar comparison of innate gene expression between microglia and astrocytes from MHV-A59-infected mice revealed that astrocytes mount delayed IFNα/β responses relative to microglia. Interestingly, the upregulation of individual IFNα/β-pathway genes in astrocytes even surpassed expression levels in microglia [46]. This suggests that astrocytes are also poor sensors of MHV to induce IFNα/β but excel as effective IFNα/β responders (Figure 1). This is supported by the ability of primary microglia, but not astrocyte cultures, to induce IFNα/β following MHV infection in vitro [53]. This inability of astrocytes to initially induce IFNα/β is most likely due to the nature of the MHV replication cycle, as heterologous neurotropic infections associated with both productive and abortive replication in astrocytes elicit IFNα/β [21,54,55]. The crucial role of IFNAR signaling in astrocytes is further evident in GFAPcre/IFNAR^fl/fl^ mice infected with MHV-A59, where mice lacking IFNAR in astrocytes did not limit virus dissemination or control the infection, resulting in mortality by day 7 p.i. [46].

Neurons are highly vulnerable targets for many neurotropic infections [56,57]. Despite their lower basal expression levels of genes in the IFNα/β signaling pathway compared to glia [20], they are sufficient for the early control of some viral infections [18,58,59,60]. Region-specific innate responses in neurons may further determine susceptibility to viral infection [61]. The limited production of IFNα/β by neurons in response to viral and prion infections [59,61,62,63,64] may be attributed to its potential toxicity to neurons [26]. Nevertheless, by responding to IFNα/β, neurons may be sufficiently primed to activate effective anti-viral molecules or proinflammatory chemokines [16]. This is supported by MHV-A59 infection of CaMKIIcre/IFNAR^fl/fl^ mice, in which forebrain neurons are deficient in IFNAR [45]. These mice succumbed to infection by day 7 p.i. due to uncontrolled virus dissemination not only in neurons but also in glia. Interestingly, higher levels of phosphorylated Signaling Transducer and Activator of Transcription 1 (pSTAT1) in neurons compared to microglia in both control IFNAR^fl/fl^ and CaMKIIcre/IFNAR^fl/fl^ mice indicated the active participation of neurons in IFN signaling. However, as IFNγ was more highly expressed in CaMKIIcre/IFNAR^fl/fl^ mice relative to controls, and pSTAT is activated by both IFNAR and IFNγR signaling, it was unclear which cytokine activated STAT1 in control mice [45].

Although IFNAR abrogation in astrocytes and neurons resulted in uncontrolled MHV spread and mortality, overall acute IFNα/β or early proinflammatory responses within the CNS remained largely intact [45,46]. This indicates that other CNS resident cells and infiltrating innate immune cells contribute to the early anti-viral responses, but that virus replication outruns both the otherwise protective innate and subsequent adaptive responses. Slightly delayed mortality compared to global IFNAR^−/−^ mice supports this notion [44]. Collectively, these findings emphasize the critical role of IFNAR responsiveness in both astrocytes and neurons in thwarting viral dissemination and protecting from acute MHV-induced mortality. The overall findings of IFNα/β responses following MHV infection are illustrated in Figure 1.

## 3. Crosstalk between IFNα/β and IFNγ

While IFNα/β is essential in limiting viral spread through the CNS, IFNγ and, to a lesser extent, perforin are crucial T cell-derived mediators reducing infectious virus to undetectable levels [32,65]. Both types of IFNs use distinct receptors but share downstream signaling components and can induce a vast set of overlapping ISG [66]. IFNα/β shapes the transition from innate to adaptive responses by activating DCs and enhancing antigen (Ag) processing and presentation by MHC-I molecules, thereby facilitating activation and expansion of Ag-specific T cells, which then migrate to the target tissue to exert anti-viral activity upon recognition of cognate Ag [8,66,67,68]. CNS resident glia and neurons express sparse MHC-I on their cell surface but can upregulate components in the MHCI-I antigen processing pathway, including MHC-I heavy chains in response to both IFNα/β and IFNγ [69,70]. However, similar to overexuberant innate responses, unchecked CD8 T cell responses may lead to toxicity and extensive tissue damage. The IFNα/β response incorporates negative regulators, such as suppressors of cytokine signaling (SOCSs) or programmed death-ligand 1 (PD-L1) and PD-L2, which downregulate ongoing activation and control excessive T cell function [71,72,73,74]. As both Ag-loaded MHC I and inhibitory molecules can be upregulated by IFNα/β or IFNγ, the net result of susceptibility to T cell function is, in part, determined by cell-specific responsiveness to either cytokine, in addition to basal expression levels.

Analysis of microglia and oligodendrocytes derived from MHV-JHM v2.2-1-infected mice revealed that oligodendrocytes exhibit delayed MHC-I surface expression compared to microglia [49]. Transcription patterns of genes encoding MHC-I heavy chains and Ag processing components were distinct both at basal levels and following infection. Microglia from naïve mice expressed high levels of these mRNAs, whereas they were near detection limits in oligodendrocytes; however, the relative increase following infection was more robust in oligodendrocytes. Further, MHC-I upregulation was IFNγ-dependent in oligodendrocytes, whereas IFNα/β signaling was sufficient for microglia MHC-I upregulation. A stricter dependence of oligodendrocytes on IFNγ was also noted for the upregulation of PD-L1 [52]. This strict control of oligodendrocytes in engaging CD8 T cells may serve to limit anti-viral CD8 T cell effector activity and pathology. The absence of PD-L1 indeed leads to the enhanced control of infectious virus at the cost of more severe axonal damage [75]. It remains to be determined whether the more limited responsiveness of oligodendrocytes to IFNα/β relative to microglia may facilitate and enhance IFNγ responses due to the enhanced availability of common signaling molecules, e.g., STAT1 and reduced expression of IFNα/β-induced inhibitory factors.

A clear link in the cross-regulation of IFNα/β and IFNγ is also revealed by MHV-A59 infection in mice with conditional deletion of IFNAR in astrocytes (GFAPcre/IFNAR^fl/fl^) or neurons (CaMKII cre/IFNAR^fl/fl^) [45,46]. Uncontrolled replication in the CNS of these mice was not associated with overall impaired IFNα/β responses, T cell infiltration, IFNγ production by T cells, or overall IFNγ protein levels. The apparent inability of IFNγ to diminish virus CNS loads suggested compromised IFNγ signaling. Indeed, microglia and infiltrating macrophages did not upregulate MHC-II surface expression, consistent with the failure to induce transcription of the IFNγ dependent class II transactivator (CIITA), the master regulator of MHC-II expression. The idea that elevated IFNα/β in infected GFAPcre/IFNAR^fl/fl^ mice relative to control mice causes dysregulated IFNγ responsiveness is supported by earlier studies indicating that heightened IFNα/β responses can act as a negative regulator of IFNγ signaling [76,77,78]. The in vitro activation of primary macrophages with poly I:C to induce IFNα/β followed by IFNγ exposure confirmed inhibition of IFNγ responsiveness [46]. As noted above, exposure of myeloid cells to increased virus-induced IFNα/β in GFAPcre/IFNAR^fl/fl^ mice may activate downstream signaling pathways shared with IFNγ signaling, thereby counteracting the IFNγ response. Negative regulators such as SOCS and ubiquitin-specific peptides 18 (USP18) may also inhibit downstream IFNγ receptor signaling [72]. Paradoxically, however, under the selective IFNAR depletion conditions, IFNγ signaling and anti-viral activity should be enhanced in cell types devoid of IFNAR. Whether impaired IFNγ signaling is limited to myeloid cells and the cell-type-specific thresholds for dysregulated IFNγ responses by IFNα/β are thus clear questions for future investigation. Similarly, the mechanisms underlying this cross-regulation may involve more complex regulation due to altered cell metabolism and mitochondrial function [79,80,81].

In summary, the IFNα/β-dependent sensitization of CNS cells to subsequent IFNγ responses may provide a mechanism to limit excessive T cell immune responses that can lead to tissue damage. Although tight control of both IFNα/β and IFNγ responses at the individual level is well recognized, a better understanding of the cross-talk between IFNα/β and IFNγ responses in vivo may offer additional cues to protect the host from exacerbated proinflammatory reaction.

## 4. Anti-Viral and Immune Modulatory Roles of Select Innate Molecules

The importance of IFNα/β induction and signaling for viral control and host survival is clearly evident from studies in global and conditional IFNAR-deficient mice, as discussed above. Understanding how individual innate immune factors contribute to the detection of infection and exert protective anti-viral functions in vivo is also emerging.

### 4.1. MDA5 and MyD88

In vitro and in vivo studies revealed that MHV-A59 induces IFNα/β via the cytoplasmic PRR, MDA5 [51]. However, signaling through myeloid differentiation primary response 88 (MyD88), an adapter protein transmitting signals from toll-like receptors (TLRs), IL-18R and IL-1R [82,83], also contributes significantly to the early induction of IFNα/β (Figure 2) and numerous proinflammatory factors following MHV-JHM v2.2-1 infection [84]. The defects in early IFNα/β and select proinflammatory factors in the absence of MyD88 were rapidly overcome. However, myeloid and CD4 T cell recruitment into the CNS, as well as CD4 T cell IFNγ production, were significantly blunted. By contrast, CD8 T cell recruitment and CD8 T cell IFNγ production were not affected. As a result, MyD88 deficiency resulted in ineffective virus control, worse clinical scores, and more severe demyelination. The results highlight a crucial role for MyD88 in accelerating and elevating not only innate immune responses but also promoting protective CD4 T cell activation. It remains to be established to what extent the protective effect of MyD88 is mediated through TLRs, IL-18R, and/or IL-1R activation. Both IL-1β and IL-18 are cleaved from inactive precursors and secreted following the activation of inflammasomes, a process involving the activation of caspase-1 and -11 [85]. Signaling through their respective receptors regulates numerous cells in both the innate and adaptive immune system, imprinting the subsequent immune responses [86]. MHV-A59 CNS infection of mice deficient in caspase-1 and -11, thereby lacking inflammasome signaling, showed poor survival and elevated viral replication compared to WT controls [87]. Efforts to delineate IL-1 from IL-18 signaling using respective receptor-deficient mice revealed that the absence of either IL-1R or IL-18R resulted in elevated viral replication; however, mortality was only increased in the absence of IL-18 signaling. Furthermore, the protective effect of IL-18 was attributed to elevated IFNγ levels in serum and spleens, which was supported by a higher percentage of splenic CD44^+^ CD4^+^ T cells producing IFNγ. T-cell function in the CNS was not assessed. These results supported that inflammasome signaling provides protection from MHV-A59 infection in part via the pro-inflammatory effects of IL-18 in supporting IFNγ production. The data are also complementary to the Myd88-mediated promotion of IFNγ production.

### 4.2. PKR

Among the many ISGs upregulated by MHV infection that have been characterized in vivo are double-stranded RNA-dependent protein kinase (PKR), 2′,5′-oligoadenylate synthetase (OAS)2, and members of the IFN-induced proteins with tetratricopeptide repeats (IFIT) family (Figure 2). The activation of PKR can regulate numerous functions, including anti-viral activity, immune responses, apoptosis, and neurotoxicity [88]. MHV-JHM v2.2-1 infection upregulates *Pkr* mRNA coincident with *Ifnα/β*; however, unlike transient upregulation of *Ifnα/β* transcripts, *Pkr* mRNA is sustained throughout T cell-mediated viral control [89]. Activated PKR was expressed in both infected and neighboring, uninfected cells. However, the infection of PKR^−/−^ mice did not impair *Ifnα/β*, *Ifit1*, or *Ifit2* mRNA upregulation. Furthermore, virus was only modestly increased during the innate infection phase, but anti-viral T cell control remained intact. The absence of PKR also did not affect CNS IL-1β, CCL5, or CXCL10 production despite significantly reduced levels of the respective mRNAs. Surprisingly, infected PKR^−/−^ mice revealed novel positive regulatory effects of PKR on tissue inhibitor of metalloproteinases inhibitor 1 (*Timp-1*), *Il-21*, and *Il-10* mRNA expression, all prominently associated with CD4 T cells during MHV-JHM v2.2-1 encephalomyelitis. Impaired *Timp-1* and *Il-21* expression correlated well with increased numbers of T cells in the parenchyma and decreased *Ighg* mRNA levels in the brain. Nevertheless, the resulting imbalance between pro- and anti-inflammatory responses in the CNS did not affect the severity or progression of clinical disease or tissue damage. Overall, the data highlight PKR-mediated immune modulation of key pro- as well as anti-inflammatory molecules controlling neuroinflammation without apparent effects on virus replication. A limited effect on virus replication is supported by only modestly reduced replication of MHV-A59 in WT compared to PKR^−/−^ macrophage cultures [90].

### 4.3. OAS/RNase L

Similar to PKR, the OAS/RNase L pathway exerts activity at multiple levels, including the degradation of viral and host RNA, induction of apoptosis, and propagation of the IFNα/β pathway through RNA degradation intermediates (Figure 2). Latent RNaseL monomers are activated by 2′-5′-linked oligoadenylates (2-5A), which are unstable products generated from adenosine triphosphate (ATP) upon the activation of the OAS family of proteins by viral double-stranded RNA [91,92]. Activated RNase L, in turn, cleaves single-stranded RNA 3′ of U-A and U-U sites found in both viral and cellular RNA. RNase L endoribonuclease activity is thus not limited to viral RNA but also degrades host cell RNA, which can, in turn, induce IFNβ and amplify the innate response [93]. As OAS, but not RNaseL genes, are ISGs, elevated OAS levels enhance the anti-viral state via RNaseL activation. It is critical to note that the inappropriate activation of RNase L is counterbalanced by the unstable nature of 2′-5′A oligomers and the endogenous RNase L inhibitor, RLI [94].

The infection of RNaseL^−/−^ mice with MHV-JHM v2.2-1 surprisingly showed no effect on overall viral control or *Ifnα/β* mRNA expression in the CNS [95]. The susceptibility to increased morbidity and mortality by two weeks p.i. could also not be attributed to altered proinflammatory signals or composition of cells infiltrating the CNS. However, histological analysis revealed that RNaseL specifically protected the brain stem from sustained infection and prevented the spread of virus to microglia/macrophages located in spinal cord grey matter. There was no evidence for enhanced neuronal infection. This subtle regional alteration in viral tropism in the absence of RNaseL coincided with increased apoptotic cells and earlier disease onset and significantly increased the severity of axonal damage and demyelination. The enhanced propensity for macrophage infection was subsequently explained by the cleavage of 2-5A via the 2′,5′-phosphodiesterase activity exerted by the MHV nonstructural protein 2 (ns2) [90]. The analysis of the replication capacity of an ns2 mutant virus expressing an inactive enzyme revealed inefficient replication in macrophages compared to WT virus, as 2-5A levels were sufficient to trigger RNaseL anti-viral activity. Prior in vivo studies had already shown a requirement for ns2 for efficient replication in the liver but not the brain [96]. The organ- and cell-type-specific effects of nsp2 antagonism on IFN-induced anti-viral activity were subsequently attributed to cell-type-specific basal and inducible expression levels of OAS genes [97]. The ns2 mutant virus replicated with similar kinetics as WT virus in primary neurons, astrocytes, and oligodendrocytes, but exhibited reduced replication in microglia and macrophages. This was consistent with the activation of RNaseL only by the mutant virus in the myeloid cells. A mechanistic investigation revealed that microglia expressed the highest basal *Oas* mRNA levels compared to the other glia and neurons and induced the highest transcript levels following infection. This correlated with the select ability of microglia to induce IFNβ in response to MHV infection. Nevertheless, although IFNα/β induction by poly I:C or direct exogenous IFNα/β treatment induced elevated *Oas* mRNAs in astrocytes, oligodendrocytes, and neurons, RNaseL was not activated by infection. This cell-type-specific activation of RNaseL may reside in distinct cellular replication niches determining viral RNA accessible to PRR recognition. Irrespectively, the protective role for RNaseL in MHV-JHM v2.2-1 CNS encephalomyelitis independent of overt anti-viral activity provides another example of how pathways induced by ISG upregulation in the CNS may severely affect the balance between neuroprotection and neurotoxicity.

### 4.4. IFIT2

Distinct from PKR and RNaseL, the Ifit2 member of the Ifit group of ISGs exerts potent anti-viral activity in the CNS following MHV-A59 infection. *Ifit2* mRNA is amongst the most highly upregulated ISG during many experimental CNS infections and is crucial to limit not only MHV-A59 but also WNV, VSV, and Rabies virus infection [50,98,99,100,101,102,103,104]. Intriguingly, the anti-viral activities of Ifit2 appeared to be neuronal tissue-specific, as demonstrated by the selective effects of Ifit2 deficiency only following VSV infection of the CNS but not footpads [105]. Both human and murine IFIT2/Ifit2 proteins can interact with multiple cellular and viral RNAs as well as cellular proteins, including microtubules, consistent with the highly pleiotropic effects on modulating host immunity [98,103,106]. IFITs are also currently studied in malignancies, where they appear to regulate cancer cell migratory activity [107,108,109].

MHV-A59 CNS infection induces strong upregulation of *Ifit2* transcripts in CNS infiltrating macrophages and, to a lesser extent, in microglia, oligodendrocytes, and astrocytes. Ifit2 protein was prominently detected in myeloid cells and neurons [50]. Infected Ifit2^−/−^ mice exhibit viral dose-dependent mortality. Uncontrolled virus replication within the CNS coincided with reduced induction of *Ifnα/β* and ISGs transcripts. Impaired IFNα/β in the absence of Ifit2 was recapitulated in in vitro infected primary macrophages and attributed to deficient IRF3 phosphorylation [50]. Subsequent studies using infection with RSA59, an isogenic spike protein recombinant EGFP expressing MHV-A59, intriguingly revealed that Ifit2^−/−^ microglia retained their ramified morphology and did not form microglia nodules early during infection [100]. The absence of Ifit2 was also associated with the downregulation of the chemokine receptor CX3CR1 expression on microglia, thereby potentially altering functions mediated by its ligand, CX3CR1. CX3CR1 is expressed by activated endothelial cells as well as neurons to maintain microglia in a quiescent state. How the dysregulation of this axis may affect the functions of endothelial cells, neuronal cells, or microglia remains to be determined. RSA59-infected Ifit2^−/−^ mice further showed a decrease in CNS accumulation of NK cells, CD4 T cells, and, to a lesser extent, CD8 T cells, although *Ifnγ* mRNA levels were increased [100]. Reduced T cells in the CNS during acute infection correlated with impaired T cell activation in draining cervical lymph nodes (cLN) and an intact blood–brain barrier (BBB) integrity compared to WT mice [101]. Finally, although Ifit2^−/−^ mice survived with a low RSA59 inoculating dose until day 30 p.i., an abundance of persisting viral protein resulted in severe white matter demyelination and clinical disease [101]. Overall, studies in MHV-A59-infected Ifit2^−/−^ mice imply that Ifit2 acts as a positive regulator of virus control, IFNα/β signaling [50], microglia activation, the accumulation of CNS lymphocytes [100], the activation of T cells in cLN, and the limiting of demyelination [101].

Interestingly, Ifit2^−/−^ mice also exhibit enhanced susceptibility to experimental autoimmune encephalomyelitis (EAE), characterized by increased demyelination and impaired uptake of damaged myelin [110]. However, contrasting the virus studies, the EAE study revealed exacerbated microglial activation, immune cell CNS infiltration, and enhanced proinflammatory gene expression coincident with metabolic reprograming towards an elevated glycolytic pathway. The apparently opposing results in the virus and EAE model highlight a context-dependent role of Ifit2 in regulating multiple immune functions, which may reside in the distinct nature of immune stimulation and the stages of CNS inflammation being studied.

Results from the MHV infection models thus demonstrate how a single ISG can regulate multiple innate as well as adaptive immune functions. The contribution and mechanism of RNA versus protein binding activities of Ifit2 in individual cell types will be critical to dissect in future studies. Specifically, the IFIT2/Ifit2-mediated regulation of many processes involving microtubules, including apoptosis, proinflammatory signatures, phagocytosis, metabolism, and cancer cell migration, beckons further investigation into its interactions with microtubule proteins [98,106,107,108,109,110]. Microtubule organization and dynamics are especially important in the CNS, where they not only regulate functions in neurons and axons but also in oligodendrocytes, astrocytes, and microglia, which all exist in a highly ramified state during homeostasis [111]. Given the reliance of many resident CNS cell types on microtubule dynamics, a potential role of Ifit2 in altering homeostatic stabilized microtubule skeleton may be central for its protective functions in multiple virus infections.

## 5. Participation of Cells Promoting T Cell Access to the Parenchyma

Stromal cells at meningeal and perivascular sites, along with myeloid cells, play a crucial role in controlling MHV within the CNS despite not being directly associated with IFNAR signaling and anti-viral pathways. Below, we will specifically discuss the roles of stromal cells producing lymphoid chemokines and myeloid cells in the enhancement of T cell access to the parenchyma. More extensively studied chemokine/chemokine receptor pathways contributing to viral control have been comprehensively reviewed [112,113,114,115].

### 5.1. Meningeal Stromal Cell Activation in Promoting Protective CD8 T Cell Immunity

Peripheral immune cells recruited to the CNS via the bloodstream initially congregate in the perivascular space between the abluminal side of endothelial cells and astrocyte endfeet. Their migration into the parenchyma is guided by regional cues such as chemokines, integrins, and the breakdown of extracellular matrix. In addition to the endothelial cells and astrocytes, additional cellular components of the perivascular unit regulating leukocyte migration include perivascular fibroblasts, pericytes, and microglia [116,117]. Fibroblastic stromal cell activation has received significant attention based on the ability of these cells to provide a scaffold for ectopic lymphoid follicle formation and promote local activation of T and B cells contributing to pathology [118,119] While a network of fibroblastic stromal cells has been noted during MHV infection [120], their direct participation in effective CD8 T cell immunity was first revealed by Cupovic et al. [121]. These studies showed that acute MHV-A59 i.n. infection induces the production of the lymphoid chemokines CCL19 and CCL21, which guide CCR7^+^ T cell localization in lymphoid tissues [122]. Importantly, the only cells producing CCL19 were endothelial cells and perivascular fibroblasts, contrasting the detection of CCL19 production from astrocytes [123] or microglia [124,125] in other neuroinflammation models. A combination of adoptive transfer approaches of virus-specific T cell receptor (TCR) transgenic T cells expressing or lacking CCR7 in infected recipient mice deficient in the chemokines or the CCR7 receptor revealed that CNS stromal cell-derived CCL19 and CCL21 were necessary for the recruitment of CD8^+^ but not CD4^+^ T cells. Furthermore, the absence of this pathway resulted in inefficient virus control and mortality. CCR7-dependent T cell migration into or retention in the CNS had been previously defined in other neuroinflammatory models [124,126], but CNS perivascular fibroblasts as a source of chemokines were novel. Endothelial cells and perivascular fibroblasts, in response to MHV infection, also upregulated the expression of cell adhesion molecules, including intracellular adhesion molecule 1 (ICAM1), vascular cell adhesion molecule 1 (VCAM1), and CD44 along with MHC-I. This suggests their potential contribution to T cell retention and activation, respectively. The potential role of CCR7^+^ migratory DCs in engaging CD8^+^ T cells in cLN or perivascular sites during inflammation [127] was not assessed. However, the results clearly supported a critical role of stromal cell activation and the expression of lymphoid chemokines CCL19 and CCL21 in specifically regulating CD8^+^ T cell migration and anti-viral effector function.

More investigation into the cellular sources and signals governing the initial production of lymphoid chemokines and their cessation is highly important to better understand potential benefits in their activation for antiviral immunity but also inhibit their ongoing activation associated with pathogenic ectopic follicles.

### 5.2. Role of Myeloid Cells in Regulating T Cell Parenchymal Access and Function

Microglia activation is a hallmark of viral infection associated with numerous protective but also deleterious effects [128,129]. Furthermore, monocyte-derived macrophages comprise early accumulating innate cells following infection. The MHV infection model has demonstrated distinct signatures of these populations both at the bulk and single-cell levels [130,131]. While much interest has focused on the role of these populations in demyelination and repair, they also play distinct roles in promoting T cell immunity, as summarized below.

Several techniques have been used to parse out the roles of microglia versus macrophages in MHV encephalomyelitis, including Ccr2^−/−^ and Ccl2^−/−^ mice, in which monocyte recruitment is impaired, and the administration of PLX5622, an inhibitor of colony-stimulating factor 1 receptor (CSF1R) tyrosine kinase. Although PLX5622 mostly depletes microglia, a caveat is that it also has limited effects on peripheral immune cells [132,133]. The PLX5622 treatment of MHV-JHM v2.2-1-infected mice prior to and coincident with infection resulted in higher viral load and increased mortality. Importantly, the delayed depletion of microglia starting at day 6 p.i. had no effect on mortality [134]. These data indicated microglia play an early acute role in initiating a protective immune response. Surprisingly, inflammatory molecules known to be produced by microglia, such as IFNs and IL6, were not affected in PLX5622-treated mice, suggesting microglia are not the sole producers. A compensatory effect for the lack of microglia was evident by increased numbers of macrophages at day 6 p.i. expressing modestly more *Ccl4* and *Ccl5* mRNA [134], both of which are ligands for CCR5 expressed by microglia, T cells, and macrophages. However, both overall and virus-specific CD4^+^ T cells were decreased, with no changes in CD8^+^ T cell accumulation. These results show that macrophages, in part, compensate for the lack of microglia but that microglia acting acutely are integral to recruiting immune cells and controlling infection.

A separate study which also used CSF1R antagonism to deplete microglia prior to infection confirmed the critical role of microglia in controlling infection and overall survival [135]. This study found an increase in CD8^+^ T cell infiltration in microglia-depleted mice but no change in CD4^+^ T cell numbers. Interestingly, both CD4^+^ and CD8^+^ T cell subsets showed reduced activation, as indicated by lower *Cd69* and *Cd44* mRNA expression in CD4^+^ and decreased *Tnf* transcripts in CD8^+^ T cells [135]. These results from separate groups, with overall congruent findings, establish a protective role of microglia in modulating the acute immune response by influencing T cell recruitment and/or activation to control infection and reduce mortality.

Monocyte-derived macrophages are, in part, recruited through CCL2/CCR2 signaling, and mice genetically depleted in these factors have been used to study the role of infiltrating myeloid cells in MHV neuroinflammation [136,137,138]. Impaired monocyte infiltration in Ccl2^−/−^ mice resulted in no detectable differences in CXCL10 or CCL5 chemokine production coincident with similar T cell recruitment [136]. However, T cells were restricted to the perivascular space in Ccl2^−/−^ mice, indicating that monocyte-derived macrophages provide an as yet undefined signal for T cell migration across the glial limitans into the parenchyma. The MHV-JHM v2.2-1 infection of CCR2^−/−^ mice was much more severe, resulting in a lack of virus control due to limited myeloid and T cell CNS infiltration accompanied by impaired *Ifnγ* and *Ccl5* expression [137]. Overall, these studies suggest that early infiltrating monocytes may have more subtle effects in viral control than microglia, which appear more essential for survival.

Effective anti-viral function by T cells requires their migration from the perivascular space across the glia limitans into the parenchyma. In addition to chemokines, this requires matrix metalloproteinase (MMP) activity to remodel extracellular matrix proteins. During MHV-JHM v2.2-1 infection, MMPs are expressed by both CNS resident stromal cells and infiltrating immune cells, with MMP9 predominantly expressed by neutrophils [136], MMP3 by astrocytes, and MMP12 by CD45^+^ infiltrating and CD45^−^ resident cells [139]. MHV-A59 infection also upregulated various MMPs, primarily during acute infection [140]. MMP activity is, in turn, regulated by tissue inhibitors of MMPs (TIMPs), which are known to be expressed by CD4^+^ T cells during MHV infection [139]. Only TIMP1 is increased during MHV infection, and predominantly expressed by CD4^+^ T cells. While the CD4^+^ T cell expression of *Timp1* was thought to explain the delayed migration of CD4 compared to CD8⁺ T cells into the brain parenchyma, infection of Timp1^−/−^ mice surprisingly showed increased CD4^+^ T cell retention in the perivascular space [141]. The results suggested a MMP-independent role of TIMP-1 in regulating CD4⁺ T cell access into the CNS parenchyma during acute encephalitis.

In summary, multiple cellular sources, including fibroblastic stromal cells, producing chemokines, integrins, and MMPs, govern the recruitment of anti-viral T cells to the CNS and their migration into the parenchyma. Further studies will be needed to characterize events within the meninges and perivascular space regulating leukocyte access to the parenchyma.

## 6. Conclusions and Gaps in Knowledge

The mCoV model of CNS infection has been instrumental in revealing the complex layers linking innate and adaptive immune responses and how the dysregulation of a single component can tip the balance from virus control to a lethal outcome. The rapid induction of IFNα/β and responsiveness of all virus-susceptible cells, especially cells that poorly induce IFNα/β, is critically important to stem virus dissemination throughout the CNS. Characterizing how resident CNS cells may differ in their response to any given infection is, thus, critical to better understanding evasion from host immune responses and mechanisms of incomplete virus elimination. Further, while some functions of IFNγ are clearly separated from the IFNα/β signaling, their pathways can be tightly intertwined. However, there is still a significant gap in knowledge about their cross-regulation and involvement of mitochondrial function, especially in CNS glia and neurons. While limited to very few members, the characterization of individual ISGs upregulated during mCoV infection demonstrates their participation in altering immune responses with or without overt effects on viral replication. This is illustrated by the effects of PKR activation on immune responses but not virus replication, the cell-type-specific anti-viral effects exerted by the RNaseL/OAS pathway, or the potent anti-viral activity of Ifit2 linked to multiple cellular functions. Identifying players optimizing anti-viral activity while minimizing adverse effects remains a key challenge. One molecule that may further be investigated is the 35-kDa interferon-induced protein (IFP35), which has been associated with the pathogenesis of MS [142] as well as respiratory infection caused by SARS CoV-2 and influenza virus [143]. Additionally, exploring the segregated, overlapping, and compensatory roles of microglia and monocyte/macrophages in promoting anti-viral immunity is an area of intense investigation. Finally, the participation of stromal cells, including fibroblastic cells localized in the meningeal and perivascular spaces, provides another critical link between innate and adaptive responses by upregulating lymphoid chemokines and integrins. The signals responsible for activating and deactivating this fibroblast network constitute another area of ongoing investigation as they may be more accessible to manipulation than parenchymal cells.

In conclusion, while mCoV encephalomyelitis does not represent a human CNS infection, insights from this model may reveal commonalities to other neurotropic infections, including potential relevance to long-haul COVID-19 complications. Shedding light on the ability of distinct CNS cell types to limit virus replication and interact with infiltrating leukocytes will aid in devising potential prevention or therapeutic strategies to combat diseases associated with viral CNS invasion.

## Figures and Tables

**Figure 1 viruses-15-02400-f001:**
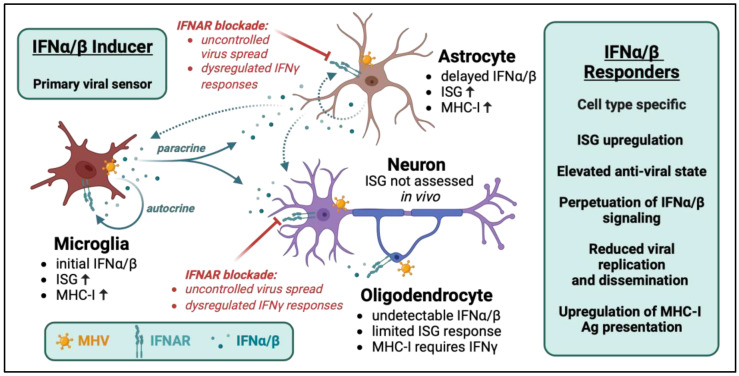
IFNα/β inducer and responder cells within the CNS following MHV infection. Microglia are initial inducers of IFNα/β, which acts in an autocrine and paracrine fashion. The increased expression of ISGs (black up arrow) elevates cellular virus sensing factors, anti-viral mediators, and MHC-I presentation (black up arrow). Astrocytes, oligodendrocytes, and neurons all respond to IFNα/β via IFNAR signaling. However, the magnitude and spectrum of ISGs expressed vary between these cell types. Astrocytes are delayed IFNα/β producers but efficiently upregulate ISGs, while oligodendrocytes are impaired in IFNα/β production and only express a limited set of genes in the IFNα/β pathway; MHC-I components are only upregulated in response to IFNγ. The ability of neurons to induce ISG in vivo has not been characterized; however, abrogation of IFNAR in both neurons or astrocytes leads to uncontrolled viral spread and mortality, emphasizing the crucial role of IFNAR signaling in both cells for protection. Lastly, while other cells compensate in amplifying IFNα/β signaling in these selectively impaired IFNAR mice and IFNγ production is not impaired, the loss of IFNγ signaling supports the idea that tight coupling between both pathways is critical to optimize their protective functions.

**Figure 2 viruses-15-02400-f002:**
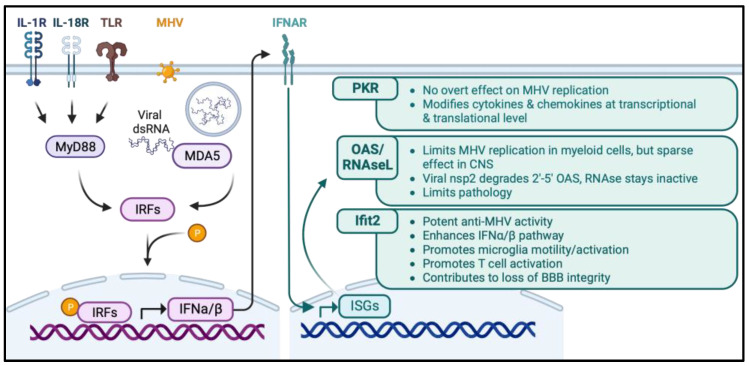
Immune-modifying and anti-viral ISGs during CNS infection by MHV. MDA5 is a primary sensor of MHV infection in microglia, leading to the induction of IFNα/β. However, signaling through MyD88, the adaptor molecule of many TLRs, IL-1R, and IL-18R, also contributes to IFNα/β production and immune activation. Among numerous ISGs upregulated in response to IFNAR signaling, PKR, OAS/RNaseL, and Ifit2 have been studied most extensively in vivo using genetically impaired mice. The cyan boxes summarize the most overt effects on viral control, immune modulation, and pathology.

## Data Availability

Not applicable.

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
