# Peer review of "The Impact of Innate Components on Viral Pathogenesis in the Neurotropic Coronavirus Encephalomyelitis Mouse Model"

_viruses, 2023, doi:10.3390/v15122400_

Round 1
Reviewer 1 Report
Comments and Suggestions for Authors
This is a comprehensive, focused review on IFNab production and responsiveness by CNS-resident and infiltrating cells to infection by neurotropic mouse coronavirus. The Bergmann lab is one of the foremost groups using this model. Balancing viral clearance against immunopathology is a theme of this review, which drives home the important point that IFNab acts as a fulcrum in this process. The timely, albeit brief, overview of FRCs in formation of ELFs is also appreciated. This reviewer learned a great deal reading this review!
(Very) Minor comments
Title – CoV should be spelled out as coronavirus, and “mouse” inserted before model
Line 41 – “in vivo” has wrong font and size
Line 115 – “aiCD4+ by” typo
Fig 1 vs text – oligodendrocytes vs oligodendroglia, pick one please
Fig 1 appears to be missing the bulletpoint consequences of IFNAR blockade for astrocytes as is shown for neurons.
Fig 1 right textbox – remove “:” after Cell type specific
Line 219 – change “is” to “are”
Refs 72-74 are reviews – can you replace these with primary literature citations?
Refs 45 and 51 are the same
Line 423 – insert space “CX3CR1is”
Line 434 – change implied to imply
Author Response
We thank you for pointing out inconsistencies and errors.
All the revisions requested were carried out and highlighted in yellow in the revised version.
We chose to use 'oligodendrocytes' throughout.
Reviewer 2 Report
Comments and Suggestions for Authors
The text is well written and complete.
The article reviews the molecular cascade secondary to viral infection, in particular that relating to the murine coronavirus.
However, no reference is made to ifi35 or IFP35.
The latter is an emerging molecule already studied in mouse models that represents an important phylogenetic link between animal and human pathology, as well as an important DAMP and ISG. Furthermore, this molecule has also been studied in the human covid-19.
Maybe it's worth mentioning. I suggest https://doi.org/10.3390/biology10121325 and https://doi.org/10.1016/j.lfs.2020.118233
Please clarify aiCD4 at line 115
Author Response
We thank the reviewer for pointing out IFP35 as a disease modifying factor. We have included a comment in the conclusions section to address this comment. We have also clarified other minor concern.